# Unsupervised Learning of Spoken Language with Visual Context

**David Harwath, Antonio Torralba, and James R. Glass**
Computer Science and Artificial Intelligence Laboratory
Massachusetts Institute of Technology
Cambridge, MA 02115
{dharwath, torralba, jrg}@csail.mit.edu

## Abstract

Humans learn to speak before they can read or write, so why can't computers do the same? In this paper, we present a deep neural network model capable of rudimentary spoken language acquisition using untranscribed audio training data, whose only supervision comes in the form of contextually relevant visual images. We describe the collection of our data comprised of over 120,000 spoken audio captions for the Places image dataset and evaluate our model on an image search and annotation task. We also provide some visualizations which suggest that our model is learning to recognize meaningful words within the caption spectrograms.

## 1 Introduction

### 1.1 Problem Statement

Conventional automatic speech recognition (ASR) is performed by highly supervised systems which utilize large amounts of training data and expert knowledge. These resources take the form of audio with parallel transcriptions for training acoustic models, collections of text for training language models, and linguist-crafted lexicons mapping words to their pronunciations. The cost of accumulating these resources is immense, so it is no surprise that very few of the more than 7,000 languages spoken across the world [1] support ASR (at the time of writing the Google Speech API supports approximately 80). This highly supervised paradigm is not the only perspective on speech processing. Glass [2] defines a spectrum of learning scenarios for speech and language systems. Highly supervised approaches place less burden on the machine learning algorithms and more on human annotators. With less annotation comes more learning difficulty, but more flexibility. At the extreme end of the spectrum, a machine would need to make do with only sensory-level inputs, the same way that humans do. It would need to infer the set of acoustic phonetic units, the subword structures such as syllables and morphs, the lexical dictionaries of words and their pronunciations, as well as higher level information about the syntactic and semantic elements of the language. While such a machine does not exist today, there has been a significant amount of research in the speech community in recent years working towards this goal. Sensor-based learning would allow a machine to acquire language by observation and interaction, and could be applied to any language universally.

In this paper, we investigate novel neural network architectures for the purpose of learning high-level semantic concepts across both audio and visual modalities. Contextually correlated streams of sensor data from multiple modalities - in this case a visual image accompanied by a spoken audio caption describing that image - are used to train networks capable of discovering patterns using otherwise unlabelled training data. For example, these networks are able to pick out instances of the spoken word "water" from within continuous speech signals and associate them with images containing bodies of water. The networks learn these associations directly from the data, without the use of conventional speech recognition, text transcriptions, or any expert linguistic knowledge whatsoever.

This represents a new direction in training neural networks that is a step closer to human learning, where the brain must utilize parallel sensory input to reason about its environment.

## 1.2 Previous Work

In recent years, there has been much work in the speech community towards developing completely unsupervised techniques that can learn the elements of a language solely from untranscribed audio data. A seminal paper in this sub-field [3] introduced Segmental Dynamic Time Warping (S-DTW), which enabled the discovery of repetitions of the same word-like units in an untranscribed audio stream. Many works followed this, some focusing on improving the computational complexity of the algorithm [4; 5], and others on applications [6; 7; 8; 9]. Other approaches have focused on inferring the lexicon of a language from strings of phonemes [10; 11], as well as inferring phone-like and higher level units directly from the speech audio itself [12; 13].

Completely separately, multimodal modeling of images and text has been an extremely popular pursuit in the machine learning field during the past decade, with many approaches focusing on accurately annotating objects and regions within images. For example, Barnard et al. [14] relied on pre-segmented and labelled images to estimate joint distributions over words and objects, while Socher [15] learned a latent meaning space covering images and words learned on non-parallel data. Other work has focused on natural language caption generation. While a large number of papers have been published on this subject, recent efforts using recurrent deep neural networks [16; 17] have generated much interest in the field. In [16], Karpathy uses a refined version of the alignment model presented in [18] to produce training exemplars for a caption-generating RNN language model that can be conditioned on visual features. Through the alignment process, a semantic embedding space containing both images and words is learned. Other works have also attempted to learn multimodal semantic embedding spaces, such as Frome et al. [19] who trained separate deep neural networks for language modeling as well as visual object classification. They then embedded the object classes into a dense word vector space with the neural network language model, and fine-tuned the visual object network to predict the embedding vectors of the words corresponding to the object classes. Fang et al. [20] also constructed a model which learned a DNN-based multimodal similarity function between images and text for the purpose of generating captions.

The work most similar to ours is Harwath and Glass [21], in which the authors attempted to associate individual words from read spoken captions to relevant regions in images. While the authors did not employ ASR to first transcribe the speech, they did use the oracle word boundaries to segment the audio caption and used a CNN to embed each word into a high dimensional space. This CNN was pretrained to perform supervised isolated word classification on a separate dataset. Additionally, the authors used an off-the-shelf RCNN object detection network [22] to segment the image into regions as well as provide embedding vectors for each region. A neural network alignment model matched the words to regions, and the resulting network was used to perform image search and annotation. In this paper, we eschew the RCNN object detection, the oracle audio caption segmentation, and the pretrained audio CNN. Instead, we present a network which is able to take as input the raw audio signal of the spoken caption and compute a similarity score against the entire image frame. The network discovers semantically meaningful words and phrases directly from the audio waveform, and is able to reliably localize them within captions. We use our network to perform image search and annotation on a new dataset of free form spoken audio captions for the Places205 dataset [23].

## 2 Data Collection

Recent work on natural language image caption generation [17; 18] have used training data comprised of parallel images and human generated text captions. There are several widely used image captioning datasets such as Flickr8k [24], Flickr30k [25], and MSCOCO [26], but the captions for these datasets are in text form. Since we desire spontaneously spoken audio captions, we collected a new corpus of captions for the Places205 dataset [23]. Places205 contains over 2.5 million images categorized into 205 different scene classes, providing a rich variety of object types in many different contexts.

To collect audio captions, we turned to Amazon's Mechanical Turk, an online service which allows requesters to post "Human Intelligence Tasks," or HITs, which anonymous workers can then complete for a small monetary compensation. We use a modified version of the Spoke JavaScript framework [27] as the basis of our audio collection HIT. Spoke is a flexible framework for creating speech-

enabled websites, acting as a wrapper around the HTML5 getUserMedia API while also supporting streaming audio from the client to a backend server via the Socket.io library. The Spoke client-side framework also includes an interface to Google's SpeechRecognition service, which can be used to provide near-instantaneous feedback to the Turker. In our Mechanical Turk collection interface, four randomly selected images are shown to the user, and a start/stop record button is paired with each image. The user is instructed to record a free-form spoken caption for each image, describing the salient objects in the scene. The backend sends the audio off to the Google speech recognition service, which returns a text hypothesis of the words spoken. Because we do not have a ground truth transcription to check against, we use the number of recognized words as a means of quality control. If the Google recognizer was able to recognize at least eight words, we accept the caption. If not, the Turker is notified in real-time that their caption cannot be accepted, and is given the option to re-record their caption. Each HIT cannot be submitted until all 4 captions have been successfully recorded. We paid the Turkers $0.03 per caption, and have to date collected approximately 120,000 captions from 1,163 unique turkers, equally sampled across the 205 Places scene categories. We plan to make our dataset publicly available in the near future.

For the experiments in this paper, we split a subset of our captions into a 114,000 utterance training set, a 2,400 utterance development set, and a 2,400 utterance testing set, covering a 27,891 word vocabulary (as specified by the Google ASR). The average caption duration was 9.5 seconds, and contained an average of 21.9 words. All the sets were randomly sampled, so many of the same speakers will appear in all three sets. We do not have ground truth text transcriptions for analysis purposes, so we use the Google speech recognition hypotheses as a proxy. Given the difficult nature of our data, these hypothesis are by no means error free. To get an idea of the error rates offered by the Google recognizer, we manually transcribed 100 randomly selected captions and found that the Google transcriptions had an estimated word error rate of 23.17%, indicating that the transcriptions are somewhat erroneous but generally reliable. To estimate the word start and end times for our analysis figures, we used Kaldi [28] to train a speech recognizer using the standard Wall Street Journal recipe, which was then used to force align the caption audio to the transcripts.

## 3  Multimodal Modeling of Images and Speech

### 3.1  Data Preprocessing

To preprocess our images we rely on the off-the-shelf VGG 16 layer network [29] pretrained on the ImageNet ILSVRC 2014 task. The mean pixel value for the VGG network is first subtracted from each image, and then we take the center 224 by 224 crop and feed it forward through the network. We discard the classification layer and take the 4096-dimensional activations of the penultimate layer to represent the input image features. We use a log mel-filterbank spectrogram to represent the spoken audio caption associated with each image. Generating the spectrogram transforms the 1-dimensional waveform into a 2-dimensional signal with both frequency and time information. We use a 25 millisecond window size and a 10 millisecond shift between consecutive frames, specifying 40 filters for the mel-scale filterbank. In order to take advantage of the additional computational efficiency offered by performing gradient computation across batched input, we force every caption spectrogram to have the same size. We do this by fixing the spectrogram size at $L$ frames (1024 to 2048 in our experiments, respectively corresponding to approximately 10 and 20 seconds of audio). We truncate any captions longer than $L$, and zero pad any shorter captions; approximately 66% of our captions were found to be 10 seconds or shorter, while 97% were under 20 seconds. It is important that the zero padding take place after any mean subtraction from the spectrograms, lest the padding bias the mean.

### 3.2  Multimodal Network Description

In its simplest sense, our model is designed to calculate a similarity score for any given image and caption pair, where the score should be high if the caption is relevant to the image and low otherwise. It is similar in spirit to previously published models which attempt to learn a similarity measure within one modality such as [30], but our model spans across multiple modalities. The specific architecture we use is illustrated in Figure 1. Essentially, we use a two branched network, with one branch devoted to modeling the image and the other devoted to modeling the spectrogram of the audio caption. The final layer of each branch outputs a vector of activations, and the dot product of

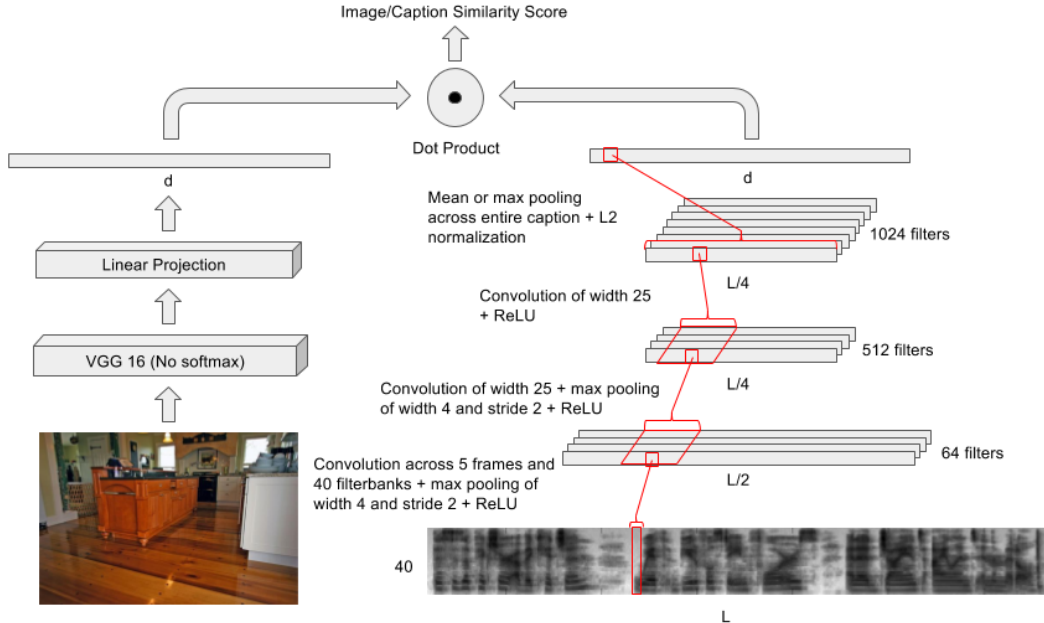

Figure 1: The architecture of our audio/visual neural network with the embedding dimension denoted by $d$ and the caption length by $L$. Separate branches of the network model the image and the audio spectrogram, and are subsequently tied together at the top level with a dot product node which calculates a similarity score for any given image and audio caption pair.

these vectors is taken to represent the similarity between the image and the caption. As described in Section 3.1, the VGG 16 layer network effectively forms the bulk of the image branch, but we need to have a means of mapping the 4096-dimensional VGG embeddings into the multimodal embedding space that the images and audio will share. For this purpose we employ a simple linear transform.

The audio branch of our network is convolutional in nature and treats the spectrogram as a 1-channel image. However, our spectrograms have a few interesting properties that differentiate them from images. While it is easy to imagine how visual objects in images can be translated along both the vertical and horizontal axes, the same is not quite true for words in spectrograms. A time delay manifests itself as a translation in the temporal (horizontal) direction, but a fixed pitch will always be mapped to the same frequency bin on the vertical axis. The same phone pronounced by two different speakers will not necessarily contain energy at exactly the same frequencies, but the physics is more complex than simply shifting the entire phone up and down the frequency axis. Following the example of [21], we size the filters of the first layer of the network to capture the entire 40-dimensional frequency axis over a context window of 5 frames, or approximately 50 milliseconds. This means that the vertical dimension is effectively collapsed out in the first layer, and so subsequent layers are only convolutional in the temporal dimension. After the final layer, we pool across the entire caption in the temporal dimension (using either mean or max pooling), and then apply L2 normalization to the caption embedding before computing the similarity score.

### 3.3 Training Procedure

By taking the dot product of an image embedding vector with an audio caption embedding vector, we obtain a similarity score $S$. We want this score to be high for ground-truth pairs, and low otherwise. We train with stochastic gradient descent using an objective function which compares the similarity scores between matched image/caption pairs and mismatched pairs. Each minibatch consists of $B$ ground truth pairs, each of which is paired with one impostor image and one impostor caption randomly sampled from the same minibatch. Let $S_j^p$ denote the similarity score between the $j$th ground truth pair, $S_j^c$ be the score between the original image and the impostor caption, and $S_j^i$ be the score between the original caption and the impostor image. The loss for the minibatch as a function

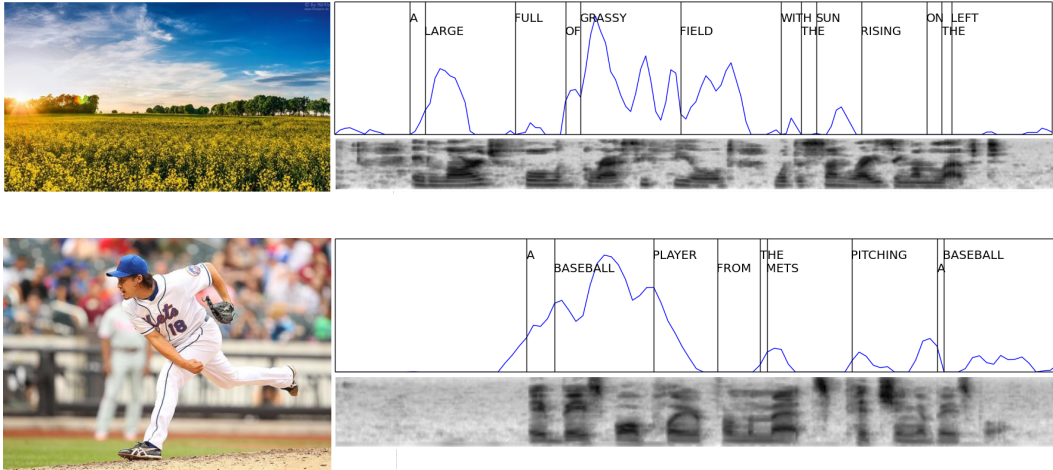

Figure 2: Examples of ground truth image/caption pairs along with the time-dependent similarity profile showing which regions of the spectrogram the model believes are highly relevant to the image. Overlaid on the similarity curve is the recognition text of the speech, along with vertical lines to denote word boundaries. Note that the neural network model had no access to these (or any) transcriptions during the training or testing phases.

of the network parameters $\theta$ is defined as:

$$\mathcal{L}(\theta) = \sum_{j=1}^{B} \max(0, S_j^c - S_j^p + 1) + \max(0, S_j^i - S_j^p + 1) \qquad (1)$$

This loss function was encourages the model to assign a higher similarity score to a ground truth image/caption pair than a mismatched pair by a margin of 1. In [18] the authors used a similar objective function to align images with text captions, but every single mismatched pair of images and captions within a minibatch was considered. Here, we only sample two negative training examples for each positive training example. In practice, we set our minibatch size to 128, used a constant momentum of 0.9, and ran SGD training for 50 epochs. Learning rates took a bit of tuning to get right. In the end, we settled on an initial value of 1e-5, and employed a schedule which decreased the learning rate by a factor between 2 and 5 every 5 to 10 epochs.

## 4 Experiments and Analysis

### 4.1 Image Query and Annotation

To objectively evaluate our models, we adopt an image search and annotation task similar to the one used by [18; 21; 31]. We subsample a validation set of 1,000 image/caption pairs from the testing set described in Section 2. To perform image search given a caption, we keep the caption fixed and use our model to compute the similarity score between the caption and each of the 1,000 images in the validation set. Image annotation works similarly, but instead the image is kept fixed and the network is tasked with finding the caption which best fits the image. Some example search and annotation results are displayed in Figures 3 and 4, and we report recall scores for the top 1, 5, and 10 hits in Table 1. We experimented with many different variations on our model architecture, including varying the number of hidden units, number of layers, filter sizes, embedding dimension, and embedding normalization schemes. We found that an embedding dimension of $d = 1024$ worked well, and that normalizing the caption embeddings prior to the similarity score computation helped. When only the acoustic embedding vectors were L2 normalized, we saw a consistent increase in performance. However, when the image embeddings were also L2 normalized (equivalent to replacing the dot product similarity with a cosine similarity), the recall scores suffered. In Table 1, we show the impact of various truncation lengths for the audio captions, as well as using a mean or max pooling scheme

"a small room which has a white piano in the corner there's a fireplace next to that and then there's a couch next to the"

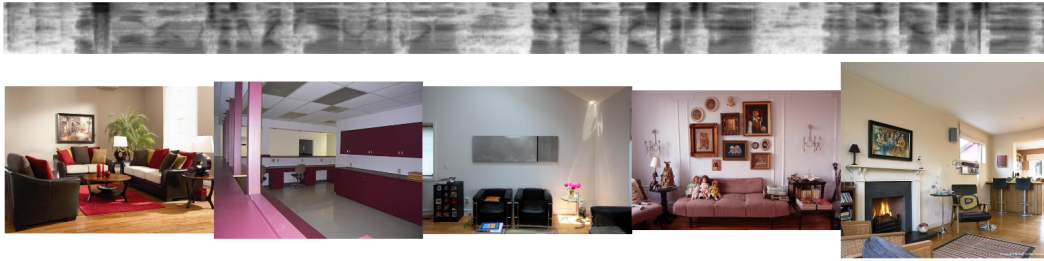

"this is a photo of a girl standing in front of a lighthouse the little girls wear blue print dress she has blonde hair and blue eyes the lighthouse"

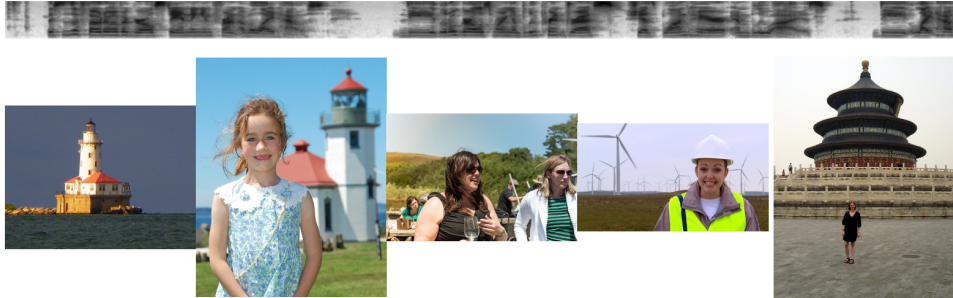

Figure 3: Example search results from our system. Shown on the top is the spectrogram of the query caption, along with its speech recognition hypothesis text. Below each caption are its five highest scoring images from the test set.

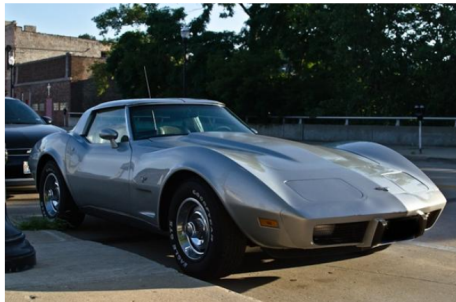

many cars are parked in the large parking lot there a large residential neighborhood with many apartment buildings

a sidewalk in front of the building there are bushes and a car parked

several green trees along a street with many parked cars

three cars are parked next to each other there's tar everywhere

car one on down the line in a factory assign sale and stop the first <spoken_noise> is

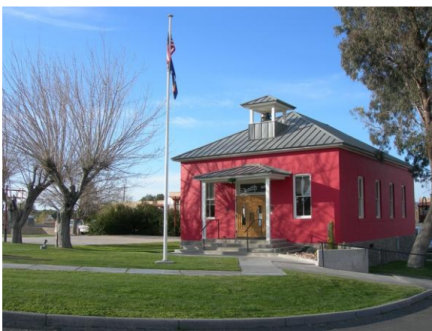

a white building with red doors and a black roof that has a tree growing up the side with red flowers

the front of an affluent home it is a ranch style house in front of the house there are several large spreading trees

this is a picture of someone's home in the blue house with white chairs in the front on the porch it also has a nice view of the street

there is a red building the red building is in front of a green lawn the lawn has been mowed recently

there's a fence in front of the house

Figure 4: Example annotation results from our system. Shown on the left is the query image, and on the right are the Google speech recognition hypotheses of the five highest scoring audio captions from the test set. We do not show the spectrograms here to avoid clutter.

| Model Variant | | Search | | | Annotation | | |
|---|---|---|---|---|---|---|---|
| Pooling type | Caption length (s) | R@1 | R@5 | R@10 | R@1 | R@5 | R@10 |
| Mean | 10 | .056 | .192 | .289 | .051 | .194 | .283 |
| Mean | 20 | .066 | .215 | .299 | .082 | .195 | .295 |
| Max | 10 | .069 | .192 | .278 | .068 | .190 | .274 |
| Max | 20 | .068 | .223 | .309 | .061 | .192 | .291 |

Table 1: Experimental results for image search and annotation on the Places audio caption data. All models shown used an embedding dimension of 1024.

across the audio caption. We found that truncating the captions to 20 seconds instead of 10 only slightly boosts the scores, and that mean and max pooling work about equally well. All models were trained on an NVIDIA Titan X GPU, which usually took about 2 days.

## 4.2 Analysis of Image-Caption Pairs

In order to gain a better understanding of what kind of acoustic patterns are being learned by our models, we computed time-dependent similarity profiles for each ground truth image/caption pair. This was done by removing the final pooling layer from the spectrogram branch of a trained model, leaving a temporal sequence of vectors reflecting the activations of the top-level convolutional units with respect to time. We computed the dot product of the image embedding vector with each of these vectors individually, rectified the signal to show only positive similarities, and then applied a 5th order median smoothing filter. We time aligned the recognition hypothesis to the spectrogram, allowing us to see exactly which words overlapped the audio regions that were highly similar to the image. Figure 2 displays several examples of these similarity curves along with the overlaid recognition text. In the majority of cases, the regions of the spectrogram which have the highest similarity to the accompanying image turn out to be highly informative words or phrases, often making explicit references to the salient objects in the image scenes. This suggests that our network is in fact learning to recognize audio patterns consistent with words using zero linguistic supervision whatsoever, and perhaps even more impressively is able to learn their semantics.

## 4.3 Analysis of Learned Acoustic Representations

To further examine the high-level acoustic representations learned by our networks, we extracted spectrograms for 1645 instances of 14 different ground truth words from the development set by force aligning the Google recognizer hypotheses to the audio. We did a forward pass of each of these individual words through the audio branch of our network, leaving us with an embedding vector for each spoken word instance. We performed t-SNE [32] analysis on these points, shown in Figure 5. We observed that the points form pure clusters, indicating that the top-level activations of the audio network carry information which is discriminative across different words.

We also examined the acoustic representations being learned at the bottom of the audio network by using the first convolutional layer and its nonlinearity as a feature extractor for a query-by-example keyword spotting task on the TIMIT data [33]. We then concatenate delta and double delta features (in the same style as the standard MFCC39 scheme), and finally apply PCA to reduce the dimensionality of the resulting features to 60. Exemplars of 10 different keywords are selected from the TIMIT training set, and frame-by-frame dynamic time warping using the cosine distance measure is used to search for occurrences of those keywords in the TIMIT testing set. Precision@N (P@N) and equal error rate (EER) are reported as evaluation metrics. We follow exactly the same experimental setup, including the same keywords, described in detail by [9] and [12], and compare against their published results in Table 2, along with a baseline system using standard MFCC39 features. The features extracted by our network are competitive against the best previously published baseline [12] in term of P@N, while outperforming it on EER. Because [12] and [9] are unsupervised approaches trained only on the TIMIT training set, this experiment is not a completely fair comparison, but serves to demonstrate that discriminative phonetic information is indeed being

modelled by our networks, even though we do not use any conventional linguistic supervision.

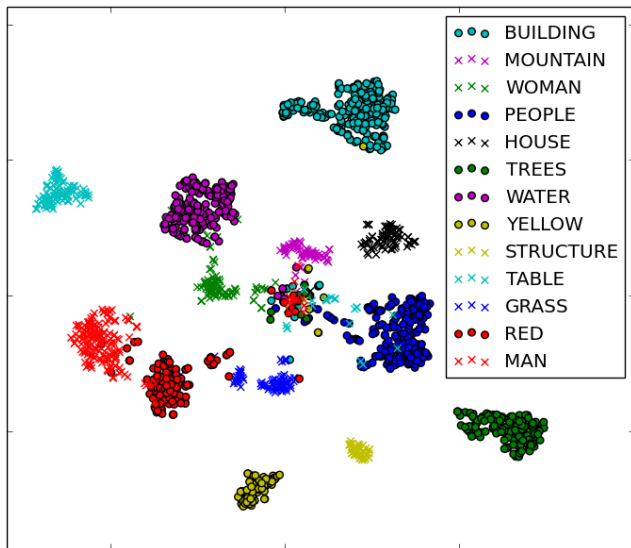

Figure 5: t-SNE visualization in 2 dimensions for 1645 spoken instances of 14 different word types taken from the development data.

| System | P@N | EER |
|---|---|---|
| MFCC baseline | 0.50 | 0.127 |
| [9] | 0.53 | 0.164 |
| [12] | 0.63 | 0.169 |
| This work | 0.62 | 0.049 |

Table 2: Precision @ N and equal error rate (EER) results for the TIMIT keyword spotting task. The 10 keywords used for the task were: *development, organizations, money, age, artists, surface, warm, year, problem, children.*

## 5 Conclusion

In this paper, we have presented a deep neural network architecture capable of learning associations between natural image scenes and accompanying free-form spoken audio captions. The networks do not rely on any form of conventional speech recognition, text transcriptions, or expert linguistic knowledge, but are able to learn to recognize semantically meaningful words and phrases at the spectral feature level. Aside from the pre-training of the off-the-shelf VGG network used in the image branch of the network, contextual information derived from the images is the only form of supervision used. We presented experimental results in which the networks were used to perform image search and annotation tasks, as well as some preliminary analysis geared towards understanding the kinds of acoustic representations are being learned by the network.

There are many possible paths that future work might take. In the near term, the embeddings learned by the networks could be used to perform acoustic segmentation and clustering, effectively learning a lexicon of word-like units. The embeddings might be combined with other forms of word clustering such as those based on dynamic time warping [3] to perform acoustically and semantically aware word clustering. Further work should also more directly explore the regions of the images which hold the highest affinity for different word or phrase-like units in the caption; this would enrich the learned lexicon of units with visual semantics. One interesting long-term idea would be to collect spoken captions for the same set of images across multiple languages and then train a network to learn words across each of the languages. By identifying which words across the languages are highly associated with the same kinds of visual objects, the network would have a means of performing speech-to-speech translation. Yet another long-term idea would be to train networks capable of synthesizing spoken captions for an arbitrary image, or alternatively synthesizing images given a spoken description. Finally, it would be possible to apply our model to more generic forms audio with visual context, such as environmental sounds.

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
