[Reviews · NeurIPS 2016]

Reviewer 1

Summary

The authors train deep neural networks to associate images and speech. They take images from the Places205 dataset and get audio descriptions from Amazon Mechanical Turk. They use a convolution-in-time speech encoder with mean pooling pretrained encoder for images (VGG) and train a final layer by using a dot product similarity with positive and negative examples. Recall rates among 1000 probes are ~5% @1, ~26% @10

Qualitative Assessment

nice idea some short cuts like fixed length of speech description good presentation of potential future work. mel not Mel why mean pooling for speech? would max pooling / kth %ile pooling work better what about pooling after the dot product? the training of a GMM ASR system is a little distracting. Make it clear that you don't pretrain the speech side of your model, or use the labels for anything except seeing how hard your data is for ASR and for labelling the spectrograms

Confidence in this Review

2-Confident (read it all; understood it all reasonably well)


Reviewer 2

Summary

This is an interesting exploration of audio-caption-to-image and image-to-audio-caption retrieval, trained with no linguistic/text supervision.

Qualitative Assessment

I think the proposed approach is reasonable (including the training architecture, training set acquisition & verification, and evaluation). The main comment I have is that the paper is rather wordy -- it seems to take the authors a while to get to their central ideas. For instance, it took a while for me to understand that fundamentally they are addressing a retrieval task, audio-to-image, and image-to-audio. Also, as I understand it, the ASR component is uniquely to vet the training data, is that correct? Other than verbosity, the writing is on the whole good, but I did notice some grammatical problems, missing prepositions etc.. In terms of originality, the authors make the contrast with reference (20), which is helpful. That work seems to use more supervision and pre-training. This brings up a limitation of their experimental analysis -- how does the largely unsupervised and trained-from-scratch proposed method compare to a method along the lines of (20)?

Confidence in this Review

2-Confident (read it all; understood it all reasonably well)


Reviewer 3

Summary

The paper presents a multi-modal neural network that learns similarities between images (or VGG objects therein) and audio captions that Turkers produced for these images. The idea is that a network would learn what "car" looks like in the audio, if the image shows a "car". While the problem of semantic "grounding" has been around for a long time, the intersection of "language and vision" and deep learning presents a new angle to this problem, which the present paper (and other works) are trying to exploit.

Qualitative Assessment

This is interesting work that is pointing into the right direction, but a few aspects of this paper are a bit problematic: 1) It would have been useful (or interesting) to use a corpus that has existing text captions, and either have users re-speak the text captions, or collect additional captions. The data collections seems generally well thought-out, but why was the Places205 data set used? Because of the focus on "objects"? Prompted speech (such as collected here) is not "spontaneous", otherwise the WSJ recognizer would not have given 20 % WER (this aspect is irrelevant for the purpose of this paper, though, I think). Typically, multiple captions are being generated for a single image. Has this been done here as well? Or is there only a single caption for each image? Why? 2) The presentation of the speech/ audio work could be a bit clearer: put all WER results (including DEL/INS/SUB) in a table, present an analysis of the most frequent words for each error type. How much data (in time and/ or words) is 100 captions? < 15 minutes? Is it possible to diagnose more clearly what the difference between the Google and Kaldi recognizers are? How big is the silence portion of the data? How was the vocabulary size (27,891) determined without manual transcription? Was there some post-segmentation of any kind to remove excess silence? (If there is any?) Which fraction of the data was clipped to 10 seconds? Which fraction was padded? Would it be possible to "sub-sample" frames, by iteratively dropping frames that are very similar to their neighbor, until 1000 frames have been reached? Is it necessary to use a CNN, rather than a simple DNN? Does it help? Has VTLN/ feature adaptation been applied? You could get those from the Kaldi recipe, I presume. (Note that the last few items are suggestions and certainly not required in the first publication in the topic) 3) The model seems a bit simplistic, although average pooling and a DNN/CNN architecture is certainly a viable first approach. I like the analysis of drawing parallels in the evaluation to see what the level of performance is with respect to other corpora and/ or tasks, but there are just too many differences. It might have been better to try more variants of the present model to gain insight into what factors influence the performance, and which ones don't (e.g. CNN vs DNN, silence segmentation approach, features, etc.). While the overall story and presentation of the work is good, the third sentence in Figure 2 contains BASEBALL twice. The first time, it is relevant (according to the similarity profile), the second time, it is not. This effect should be analyzed. Is this a feature of the model? Or an artifact that would disappear with better training etc? It is good that the authors plan to release their data (maybe even with a baseline system). The references seem good (the authors should check for other relevant work before publication, though). Finally, the paper has many small typos and grammar errors - please fix.

Confidence in this Review

3-Expert (read the paper in detail, know the area, quite certain of my opinion)


Reviewer 4

Summary

The paper presents a multimodal model to compute the similarity between images and speech on image-speech data.

Qualitative Assessment

The paper is well-written and easy to follow. The proposed model is technically sound, and the result is solid, showing the effectiveness of the model. The title is a bit misleading because the model is not really trained using unsupervised learning methods. The paired image-speech data is used for training. The training procedure described in 3.3 is a typical supervised learning case. The proposed model described in section 3 bears strong resemblance to the deep multimodal similarity model proposed in section 5 of the paper: https://arxiv.org/pdf/1411.4952v3.pdf It is necessary to discuss the paper in related work.

Confidence in this Review

2-Confident (read it all; understood it all reasonably well)


Reviewer 5

Summary

This paper proposes a neural network framework with two branches for learning spoken and visual embeddings given the pairs of images and corresponding spoken annotations. The multi-model framework does not require speech recognition and automatically learn the correlation between raw audio and visual context.

Qualitative Assessment

The motivation is interesting and can directly bridge two signal-based contents (image and speech). However, there are concerns to be addressed: 1) The related references are missing in the paper. The proposed neural net architecture is similar to the DSSM/CDSSM. The author should include [1]-[3] into the reference. The loss function used in the paper includes both sides of scores (from visual to speech and from speech to visual), [4] used the similar idea and the DSSM architecture for learning different types of embeddings. 2) In 3.3, it is not clear that why the margin is 1 in the equation and why dot product instead of cosine similarity. 3) The investigation of more experiments may be needed in order to claim the performance validation. In 4.1, because the authors did not compare the performance with the approaches that utilized the text information on the same dataset, it may not clear whether similar R@10 on different datasets is comparable or not. I would suggest the authors perform the text-based methods on the Place data to show the comparison. 4) The annotation experiment did not generate the speech based on images but retrieves the similar utterances for annotation. This can be viewed as another direction of retrieval. Therefore, it is not clear how this system (for annotation) is useful for the practical usage. The ideal way is to automatically synthesize the speech signal based on the given images, similar to image captioning. Given the concerns, more experiments should be added, where text-based approaches and synthesizing speech instead of annotation should be included in order to validate the usefulness and effectiveness. [1] Huang et al., "Learning deep structured semantic models for web search using clickthrough data," in CIKM, 2013. [2] Shen et al., "A latent semantic model with convolutional-pooling structure for information retrieval," in CIKM, 2014. [3] Fang et al., "From captions to visual concepts and back," in CVPR, 2015. [4] Chen et al., "Zero-shot learning of intent embeddings for expansion by convolutional deep structured semantic models," in ICASSP, 2016.

Confidence in this Review

3-Expert (read the paper in detail, know the area, quite certain of my opinion)


Reviewer 6

Summary

The authors propose a model which learns the similarity between spoken audio caption and images, and a dataset with spoken audio caption and images. Although the application is new to me, I think the model is similar to [1], [2], etc. in learning the similarity between two modalities. [1] Chopra, Sumit, Raia Hadsell, and Yann LeCun. "Learning a similarity metric discriminatively, with application to face verification." 2005 IEEE Computer Society Conference on Computer Vision and Pattern Recognition (CVPR'05). Vol. 1. IEEE, 2005. [2] Fang, Hao, et al. "From captions to visual concepts and back." Proceedings of the IEEE Conference on Computer Vision and Pattern Recognition. 2015.

Qualitative Assessment

Other comments: 1. The title of the paper is about unsupervised learning, but the model does utilize the paired with information between speech and images. Image features are trained in a supervised way using image labels. 2. Propose baselines should be given. To evaluate the effect of the proposed method, some standard baseline in image search, annotation. Minor errors: 1. Line 114 msny -> msny 2. Equation 1, the symbols S_i^i and S_i^p are not mentioned in the text, I think they should be S_i^I -> S_j^i; S_i^p -> S_j^p. 3. The reference should be noted using brackets, instead of parenthesis. 4. In figures 1 and 2, it will be great to show the scale of the similarity score, spectrograms.

Confidence in this Review

2-Confident (read it all; understood it all reasonably well)